# Biologically Active Tissue Factor-Bearing Larger Ectosome-Like Extracellular Vesicles in Malignant Effusions from Ovarian Cancer Patients: Correlation with Incidence of Thrombosis

**DOI:** 10.3390/ijms22020790

**Published:** 2021-01-14

**Authors:** Corinna Steidel, Fanny Ender, Achim Rody, Nikolas von Bubnoff, Frank Gieseler

**Affiliations:** 1Clinic for Hematology and Oncology, University Hospital and Medical School (UKSH), University of Luebeck, Ratzeburger Allee 160, 23538 Luebeck, Germany; nikolas.vonbubnoff@uksh.de (N.v.B.); frank.gieseler@uksh.de (F.G.); 2Department of Gynecology and Obstetrics, University Hospital and Medical School (UKSH), Ratzeburger Allee 160, 23538 Luebeck, Germany; achim.rody@uksh.de

**Keywords:** extracellular vesicles, tissue factor, ascites, thrombosis

## Abstract

The development of malignant effusions such as ascites reflects a massive progression of a malignant disease. In patients with ovarian carcinoma, a high amount of ascites (>500 mL) is an independent negative prognostic marker. The composition and constituents of ascites reflect the inflammatory environment of the underlying tumor. Increased cellular resistance of ascites-derived tumor cells and the development of venous thromboembolic events (VTE) are major risks for these patients, especially in patients with advanced ovarian carcinoma. In this study, we discuss the release of tissue factor-bearing extracellular vesicles (TF^+^ EVs) from tumor cells into the environment (ascites fluid) and their systemic spreading as a possible causal explanation of the pathologic coagulation status in these patients. We obtained ascites from patients with advanced ovarian carcinoma, collected during surgery or therapeutic paracentesis (n = 20). Larger ectosome-like EVs were isolated using sequential centrifugation, quantified by high-resolution flow cytometry and analyzed using nanoparticle tracking analysis. Furthermore, the pro-coagulant properties (TF activity) of EVs were determined. Compared to published TF activities of EVs from healthy persons, TF activities of EVs derived from ascites of patients with ovarian cancer were very high, with a median of 80 pg/mL. The rate of VTE, as reported in the patient files, was high as well (35%, 7 out of 20). Furthermore, all but one patient with VTE had EV concentrations above the median within their ascetic fluid (*p* < 0.02). Since VTE continues to be a frequent cause of death in cancer patients, prophylactic antithrombotic treatment might be worth considering in these patients. However, given the risk of bleeding, more clinical data are warranted. Although the study is too small to enable reaching a conclusion on direct clinical implementation, it can well serve as a proof of principle and a rationale to initiate a prospective clinical study with different patient subgroups. We also show ex vivo that these larger ectosome-like EVs induce intracellular ERK phosphorylation and tumor cell migration, which is not directly related to their pro-coagulative potency, but might help to understand why cancer patients with thromboembolic events have a poorer prognosis.

## 1. Introduction

The clinical association between progressive cancer and the development of venous thromboembolic events (VTE) is well known, and represents a life-threatening event in patients [1]. Patients with ovarian carcinoma are affected particularly often [2]. The molecular reasons are still unclear, and most certainly are multifactorial. Markers that might predict a higher risk would assist in the decision whether to pursue prophylactic antithrombotic treatment of these patients. As it is rich in tumor-promoting cytokines, chemokines, growth factors, proteinases, and extracellular vesicles (EVs), malignant ascites is considered a unique form of the tumor environment [3,4]. EVs are suspected of playing a major role in both cancer progression and thromboembolism, especially since it is known that they present membrane-bound tissue factor (TF), as well as phosphatidylserine (PS), thus inducing the constitution of the extrinsic tenase of the systemic coagulation system [5]. The shedding of Epithelial cell adhesion molecule positive EVs (EpCAM^+^ EVs) into the ascites of patients with ovarian carcinoma [6] has been shown previously, as well as the release of tissue-factor-positive EVs (TF^+^ EVs) from various ovarian carcinoma cell lines, especially those of the serous subtype [7]. From a clinical point of view, it would be interesting to see whether these in vitro findings can be confirmed in clinical patient material. We performed an ex vivo study using clinical material from patients with advanced and progressive ovarian carcinoma of the serous subtype (high-grade serous carcinoma, HGSC). Since the EV composition in blood, especially in the inflammatory environment of a tumor disease, is heterogeneous, and since most of the EVs in this enclosed space originate from the tumor, we decided to investigate malignant ascites from patients with ovarian carcinoma [8].

Following the suggestion of the International Society for Extracellular Vesicles (ISEV), we chose an EV nomenclature that refers to the physical characteristics (size and mass) for the experimental preparations of this study [9]. Accordingly, as we isolated a specific EV subpopulation using sequential centrifugation, including high-speed centrifugation, we designated them as high-speed EVs (hsEVs).

## 2. Results

### 2.1. Ascites from Patients with HGSC Contains High Amounts of Larger EVs

The accessibility of ascites makes it an excellent subject for the investigation of prognostic and predictive biomarkers, pharmacodynamic markers, and molecular profiling analysis [10]. It has been shown previously that ascites contains large amounts of EVs that are mostly derived from tumor cells [11]. Using sequential centrifugation, we isolated EVs from ascites of patients with HGSC and enriched hsEVs (Figure 1A) [12]. The integrity of hsEVs was confirmed by the conversion of non-fluorescent CFDA-SE into the fluorescent variant by intravesicular esterases with biological activity. Subsequently, CFSE^+^EVs were visualized and quantified by high-resolution flow cytometry using beads (Megamix) as size reference to focus on events within the size range of 0.1–0.9 µm (Figure 1B). We found a high variation of 2 log levels (min 3.8 × 10^5^, max 4.9 × 10^7^) of CFSE^+^ hsEVs in the ascites samples of the patients (Figure 1C). Additionally, we assessed the size distribution of isolated hsEVs by nanoparticle tracking analysis (NTA) and confined a selective enrichment of larger ectosome-like EVs (100–500 nm) in the pellet after high-speed centrifugation (Figure 1E). Overall, we detected hsEVs in a concentration of 6 × 10^6^ CFSE^+^ events/mL (median) in ascites from patients with HGSC.

We further confirmed the lipid bilayer structure specific for EVs by investigating the presence of the tetraspanin markers CD9, CD63, and CD81 as suggested by the ISEV [9], and found varying amounts of these markers (CD9^+^, 0.53–27.09%; CD63^+^, 1.0–12.5%; CD81, 4.18–26.25%) on the surface of hsEVs (Figure 1D). Tetraspanin markers were not uniformly present in our preparations but on a subpopulation of CFSE^+^ hsEVs, which is in line with our previous findings on isolated hsEVs from cell lines [12].

### 2.2. Elevated Levels of TF-Bearing EVs Correlate with the Incidence of Thrombosis in HGSC Patients

In our patient samples, TF activity of ascites-derived hsEVs was generally very high. TF activity varied from 0 pg/mL (min) to 509 pg/mL (max) (median = 80 pg/mL) (Figure 2A). The proportion of patients with thrombosis in their health history was high (35%, 7 out of 20) in our group of patients with progressive HGSC cancer (Figure 2A), and we found a significant association of the incidence of thrombosis reported in the patient files with the number of hsEVs: All but one (6 out of 7) of the patients with reported thrombosis events had elevated numbers of hsEVs above the median of 6 × 10^6^ hsEVs/mL (Figure 1C and Figure 2B, *p* < 0.019).

### 2.3. Ascites-Derived hsEVs are Biologically Active and Induce Intracellular Signaling and Tumor Cell Migration In Vitro

These experiments are not directly related to the topic of thrombosis-induction by TF^+^EVs, but they show that the same vesicles can have a direct influence on the progression of the cancer by activating the PAR 2/G-protein/ERK signaling pathway. For these experiments, we used the pancreas carcinoma cell line COLO 357 as an in vitro target system because these cells constantly express PAR2, which is the membrane structure that initiates the intracellular events induced by TF-associated structures of the EVs. We have previously published that TF^+^EVs are able to activate the PAR2/G-protein/ERK signaling pathway and subsequently induce tumor cell migration, which is a surrogate parameter for progression and metastization [3,13]. Here we showed that hsEVs isolated from patient ascites have the same potency, and found a significant increase of up to 184% of ERK phosphorylation in the cells after 10 min stimulation with ascites-derived hsEVs when compared to the PBS-treated controls (Figure 3) Tumor cell migration was also significantly induced compared to PBS-treated control cells (Figure 3).

## 3. Discussion

The development of ascites in patients with ovarian carcinoma indicates a massive progression of the disease, and is associated with severely reduced survival prognosis for patients [15]. In addition, these patients have a high incidence of venous thromboembolic events (VTE), which can also result in a fatal outcome [2]. In this scenario of tumor progression and activation of the coagulation system, EVs are suspected of playing a major role, especially when they expose membrane-bound TF, as well as PS, which enables them to activate the systemic coagulation system on the one hand, and the PAR2/ERK signaling pathway on the other [5,10,16].

To further explore this proposed chain of events in a clinical context, we chose ascites from patients with ovarian carcinoma as a source of EVs. The EV composition in blood, especially in the inflammatory environment of a progressive tumor disease, is much more heterogeneous than that from ascites, which is a “third space” that contains mostly EVs derived from tumor cells [11]. It is known that ascites in patients with ovarian carcinoma contains high amounts of EVs associated with TF expression, but it has only recently been shown that the detection of TF activity, which we used here, is a better parameter for coagulation activation than the antigen detection due to the existence of encrypted TF without associated activity [14].

In our study, we isolated larger ectosome-like EVs from ascites of patients with ovarian carcinoma, which we designated as high-speed EVs (hsEVs) based on the procedure of isolation (Figure 1A). Although sequential centrifugation is widely used, a complete separation of smaller exosome-like EVs and larger ectosome-like EVs cannot be achieved. Instead, larger EVs are enriched within the high-speed (hs) fraction [12]. We have shown previously that especially this subgroup of EVs is associated with TF activity [3]. In 2018, the ISEV released a position statement for minimal information for studies of EVs, which recommended a nomenclature for isolated EVs that reflects the fact that a clear separation between exosomes and ectosomes is not possible (e.g., larger ectosome-like vesicles, or hsEVs as we have used here). In addition, the ISEV suggests the use of membrane proteins that prove the existence of intact biological membranes. Therefore, we assessed the surface expression of tetraspanins (Figure 1D) and, more essential from our point of view, the conversion of CFDA-SE into the fluorescent variant CFSE through the activity of intravesicular esterases (Figure 1B). We described earlier that the proof of CFSE conversion and the presence of tetraspanins through fluorescent antibodies in high-resolution flow cytometry instead of Western blot analysis has the advantage of enabling detection of subpopulations of interest [12]. Using these methods, we determined the concentration of hsEV (Figure 1C and Figure 1B), EV-associated TF activity, and the biological effects induced by hsEVs isolated from ascites of patients with ovarian carcinoma. We found not only high hsEV numbers (Figure 1C), but also very high levels of TF activity (Figure 2A) on hsEVs from ascites samples. Hisada and coworkers proposed four response categories of EV-TF activity for platelet-poor plasma (PPP). They defined zero (0 to <0.5 pg/mL), weak (0.5 to <1.0 pg/mL), moderate (1 to <2.0 pg/mL), and strong (>2.0) groups based on their potency of FXa generation [14]. In our study, except for one preparation, ascites-derived hsEVs were all characterized by very high levels of TF activity, with a median of 80 pg/mL (Figure 2A).

The experiments showing that the same ascites-derived hsEVs are biologically active and induce intracellular signaling and tumor cell migration in vitro (Figure 3) are not directly related to the topic of thrombosis induction by TF^+^ EVs. As ascites mirrors the tumor environment, this finding can be a factor that helps to explain the fatally high resistance of tumor cells in ascites, especially since it is known that PAR2 is up-regulated during ovarian cancer progression [17].

As mentioned before, the clinical association of tumor progression and the development of potentially fatal VTE is well known, with the development of ascites being a striking of tumor progression. In the group of patients, although not selected for this event, the incidence of VTE was also very high (35%, 7 out of 20). Information on the incidence of VTE in the two years after detection of ascites was taken from the patient files as described in the methods section.

As a noticeable association, all patients with VTE belong to the group of those with hsEVs associated with high levels of TF activity above the median (Figure 2A, *p* = 0.019). Interestingly, the patient with the highest number of TF^+^ EVs did not develop thrombosis (Figure 2B), which points to the complex biochemical processes that finally result in the clinical event of thrombosis. These processes include not only the presence of TF^+^ EVs, but also direct and indirect procoagulant factors, as well as numerous anti-coagulant mechanisms [18].

This study was not designed as a prospective clinical study with the inclusion of different patient subgroups (e.g., with different prognosis groups), but it can well serve as a proof-of-principle investigation. The results suggest that TF activity of ascites-derived hsEVs can serve as a liquid biomarker for disease progression and its association with patients at risk for VTE. As described before, the low molecular weight heparin tinzaparin is able to block the TF pathway of coagulation activation by the induction of TFPI release, and thus would be a candidate to block this fatal chain of events [3]. Based on our previous work, we are in the process of initiating a prospective clinical study with the aim of defining the subgroup of patients that might benefit from prophylactic anti-thrombotic treatment.

## 4. Material and Methods

### 4.1. Characteristics of Patients, Data Acquisition, and Determination of VTE Occurrence

Between 2013 and 2017, we obtained 20 ascites samples from patients with advanced ovarian carcinoma, collected during surgery or therapeutic paracentesis in the Clinic for Gynecology of the University Hospital Schleswig-Holstein, Campus Luebeck (UKSH). The average age of patients was 68 ± 9 years (48–81 years), and all were suffering from metastasized HGSC (FIGO IIIc or IV). Histological diagnosis was determined by the Clinic of Pathology of the UKSH. In all but two patients, the ascites was the first symptom of their cancer; none of the patients received chemotherapy at the time of paracentesis. The patients had been admitted to the university hospital for paracentesis, and the probes used in our laboratory were residual material that would have been discarded otherwise. In the two years following the detection of ascites, we searched the patient files for information on acute VTE in their case histories.

The study was approved by the Ethics Committee of the University of Luebeck (file number 13-206), and all patients gave their informed consent to the examination of their ascites, the evaluation of their medical files and the publication with anonymized data.

### 4.2. Isolation of EVs from Ascites of Ovarian Cancer Patients

Ascites was centrifuged at 470× *g* for 10 min to remove residual cells, and the supernatant was frozen at −20 °C until further use. EVs were enriched by using a centrifugation protocol recommended by the International Society on Thrombosis and Haemostasis (ISTH) [19]. Briefly, samples were centrifuged at 2500× *g* for 15 min, decanted and again centrifuged for 15 min at 2500× *g*. From these samples, we isolated a specific larger (ectosome-like) subpopulation of EVs by using high-speed centrifugation at 10,000× *g* for 90 min, similar to the protocol published by Muralidharan-Chari et al. [20]. Thus, isolated EVs were enriched by larger ectosome-like vesicles and were designated as hsEVs.

### 4.3. EV Quantification by High-Resolution Flow Cytometry

Isolated hsEVs were quantified using a Novocyte high-resolution flow cytometer (ACEA Biosciences Inc., San Diego, CA, USA). As fluorescence labeling of EVs increases the detection sensitivity of flow cytometers as described before [12,21], we incubated the samples for 60 min with 40 µM carboxyfluorescein diacetate succinimidyl ester (CFDA-SE) (Cayman Chemical Company, Ann Arbor, MI, USA) in the absence of light. During that time, CFDA-SE permeated EV membranes and was enzymatically processed into the fluorescent variant CFSE, which then was detectable in the FITC channel of the flow cytometer. After incubation with CFDA-SE, samples were immediately analyzed. For evaluating the size of our hsEV population, we used Megamix Beads (BioCytex, Marseille, France) as a reference. A size gate ranging from 0.1–0.9 µm was set for quantification of hsEVs (Figure 1C).

### 4.4. Antibody Labeling of hsEVs for Flow Cytometry

To demonstrate the lipid bilayer structure of hsEVs by flow cytometry, the following antibodies binding to proteins of the tetraspanin family were used: anti-human CD9 antibody (HI9a), anti-human CD63 antibody (H5C6), and anti-human CD81 antibody (5A6), all labeled with phycoerythrin (PE) and purchased from BioLegend (San Diego, CA, USA). We used an isotype control (mouse anti-human; IgG1κ) from eBiosciences/Affymetrix (San Diego, CA, USA) in place of the primary antibody to determine the contribution of non-specific background to staining. Optimal antibody concentrations were titrated beforehand. Antibody labeling was performed for 20 min at 4 °C subsequent to CFSE labeling and just before high-speed centrifugation.

### 4.5. Nanoparticle Tracking Analysis

In this study, we used an NS300 (Malvern Instruments Ltd., Malvern, UK) equipped with a 488 nm laser module. Nanoparticle tracking analysis (NTA) visualizes and measures small particles (10–1000 nm) in suspension, based on the analysis of Brownian motion from a video sequence after being visualized by a laser beam. The scattered light of the particles was recorded with a light-sensitive camera, being arranged at a 90° angle to the irradiation plane, and then analyzed.

### 4.6. Cell Culture

To test the biological activity of isolated hsEVs, we used COLO 357 cells (European Collection of Authenticated Cell Cultures, Salisbury, UK) that were once derived from a metastasis of a pancreatic adenocarcinoma and grow as an adhering monolayer with a cell doubling time of 21 h [22]. These cells were used because of their stable expression of the protease-activated receptors PAR1 and PAR2 and their low spontaneous migratory capacity [23]. Cells were cultured under serum-free conditions using RPMI1640 (Lonza, Basel, Switzerland) supplemented with 10% panexin (Pan-Biotech, Aidenbach, Germany), 1% penicillin streptomycin with glutamine (Gibco, Thermo Fisher Scientific, Waltham, MA, USA), and 1% sodium pyruvate (Biochrom, Berlin, Germany). We substituted fetal calf serum (FCS) (Biochrom, Berlin, Germany) by panexin to avoid contaminating our experimental preparation with FCS-derived EVs. Therefore, the concentration of FCS was diminished stepwise, and panexin was increased over several weeks. Cells were seeded in cell culture flasks (Sarstedt AG & Co, Nuembrecht, Germany) and incubated at 37 °C, 5% CO_2_, and 95% humidity. Cells were free of contamination, and regular tests for detection of mycoplasma infections were performed.

### 4.7. Analysis of EV-Induced Intracellular Cell Signaling

The capability of isolated hsEVs to induce the phosphorylation of extracellular signal-regulated kinases (ERK) was assessed using the Cell-Based ERK1/2 Phosphorylation ELISA kit (RayBiotech Inc., Norcross, GA, USA). To obtain an adherent monolayer, 3 × 10^4^ cells/well were seeded and incubated overnight. Four hours before treatment, cells were subjected to starvation medium containing 1% panexin. For stimulation, cells were either incubated with PBS (PAA Laboratories GmbH, Pasching, Austria), or with isolated hsEVs from HGSC patients for 10 min at room temperature. Thereafter, cells were fixed, quenched and blocked before incubation with a murine monoclonal antihuman ERK1/2 antibody or a murine monoclonal anti-human phosphoERK1/2 antibody. Staining was performed using a secondary anti-mouse IgG antibody coupled to horseradish peroxidase (HRP). After administration of the HRP’s substrate TMB, the substrate turnover, measured at 450 nm, was proportional to the amount of phosphorylated or unphosphorylated ERK.

### 4.8. Analysis of EV-Stimulated Tumor Cell Migration

The effect of isolated hsEVs on tumor cell migration was assessed using the Oris^TM^ Pro Cell Migration Assay (Platypus Technologies LLC, Madison, WI, USA) according to the manufacturer’s instructions. Briefly, 3 × 10^4^ cells/well were seeded into a special 96-well plate equipped with a silicone stopper in the middle of each well, thus providing a cell-free detection zone. Overnight incubation ensured adherence of the cells. Afterwards, the stoppers were removed, and non-adherent cells were washed out. Cells were incubated with a mixture of 50% cell culture medium and 50% sample for 48 h, receiving a medium top-up of 100 µL in between after 24 h. Finally, cells were fixed and stained using the Diff Quick Set (Medion Grifols Diagnostics AG, Duedingen, Switzerland). The faceplate provided for the 96-well plate was applied, and the detection zone was photographed using a 5G megapixel camera (Point Grey, Richmond, BC, Canada) and an Axioskop microscope (ZEISS, Oberkochen, Germany). Images were analyzed using the Fiji Image J software. The program detected the free area by counting every pixel below a threshold. This number of pixels was subtracted from the zero value, which was determined by measuring a well that was fixed directly after removal of the stopper (time point 0). As a result, the migration area in pixel^2^ was calculated.

### 4.9. Determination of Tissue Factor Activity on the Surface of EVs

Tissue factor (TF) activity was analyzed using the Zymuphen MP-TF assay (Aniara, West Chester OH, USA) according to the manufacturer’s instruction. In short, TF-bearing hsEV were added and bound to the anti-TF-antibody-coated ELISA plate. After washing, FX and FVII were introduced into the well and incubated for 2 h. Afterwards, chromogenic FXa-substrate was added and incubated for another 2 h. Adding citric acid stopped the reaction. Finally, the OD was measured at 405 nm. Using customized standards from the same kit, the concentration of TF activity was calculated in pg/mL.

### 4.10. Statistical Analysis

To ensure low intra-experimental variation, experiments were performed at least in triplicate. Mean and standard deviation of the individual values were determined accordingly. Furthermore, the median was calculated from the individual values and expressed as box plots that included the upper and lower quartiles, as well as the extreme values. To evaluate the correlation between the absolute amount of hsEVs/mL ascites and the incidence of thrombosis, we used the chi-square test. Statistical analysis was performed using GraphPad Prism version 8.2.1. (San Diego, CA, USA). *p* values < 0.05 were considered statistically significant.

## Figures and Tables

**Figure 1 ijms-22-00790-f001:**
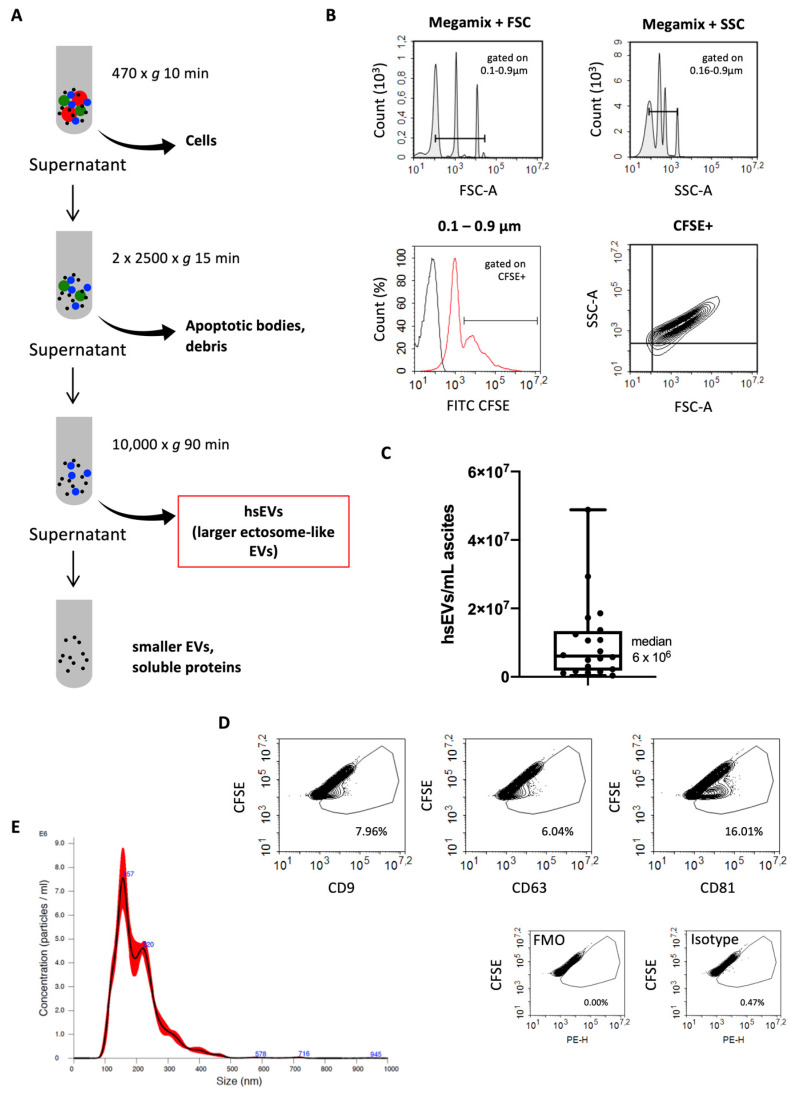
Ascites from patients with ovarian cancer contains high amounts of larger EVs. (**A**) Purification of EV subsets using sequential centrifugation. (**B**) Gating strategy for the identification of intact high-speed EVs (hsEVs) by high-resolution flow cytometry. A gate representing the size range of interest (0.1–0.9 µm) was set according to known diameters of standard beads (upper plots). Within the estimated size range of 0.1–0.9 µm, carboxyfluorescein diacetate succinimidyl ester (CFSE^+^) events were identified as intact hsEVs (lower plots). For setting the CFSE gate, an unstained PBS control (black histogram) and a stained hsEV sample (red histogram) were overlaid. Of note, the intermediate signal occurring in the stained hsEV sample can be assigned to CFSE noise, as it has been recently shown to be EV independent [12]. (**C**) Concentration of CFSE^+^ hsEVs/mL isolated from ascites of ovarian carcinoma patients expressed as a box plot including median (line), upper and lower quartiles (box), and extreme values (whiskers). (**D**) Flow cytometry analysis of tetraspanin markers CD9, CD63, and CD81 on hsEVs (upper panel), fluorescence minus one (FMO) control (lower panel left), and isotype control (lower panel right). Data shown are representative of five independent experiments. (**E**) Analysis of hsEV size distribution by nanoparticle tracking analysis (NTA). Shown is the mean (black line) ± SD (red line) from three recordings of the same sample. Representative picture from the analysis of 5 different hsEV preparations.

**Figure 2 ijms-22-00790-f002:**
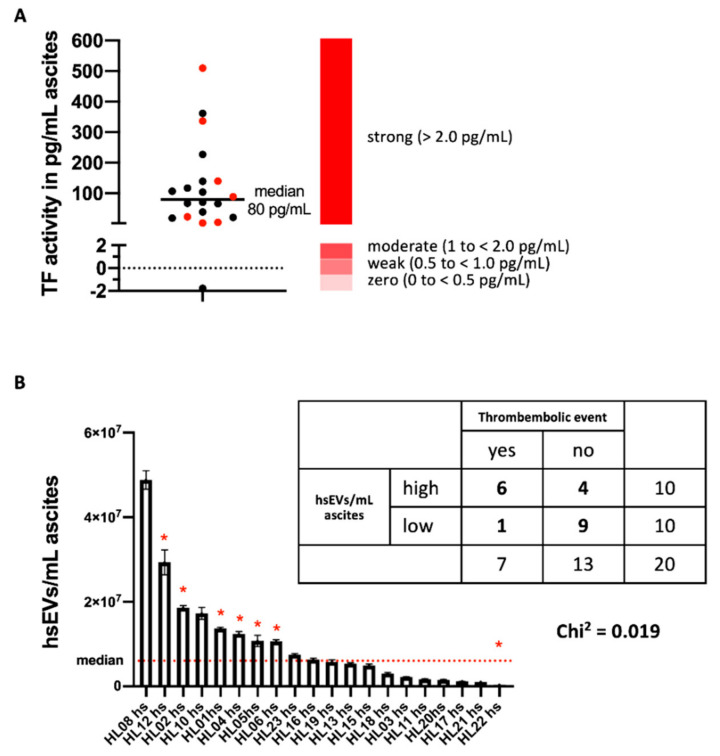
Elevated levels of TF-bearing EVs correlate with the incidence of thrombosis in ovarian cancer patients. (**A**) Tissue-factor (TF) activity on ascites-derived hsEVs isolated from ovarian carcinoma patients. The median of 20 individual samples (80 pg/mL) is shown. The four response categories of EV-TF activity proposed by Hisada and coworkers served as reference [14] (right, scale in red). (**B**) Quantification of ascites-derived hsEVs isolated from ovarian carcinoma patients in the size range of 0.1–0.9 μm. The median of 6 × 10^6^ hsEVs/mL ascites (depicted by the red dotted line) is implemented. The correlation between the absolute amount of hsEVs/mL ascites and the incidence of thrombosis was analyzed using the chi-square test (* *p* = 0.019) (see table on the right).

**Figure 3 ijms-22-00790-f003:**
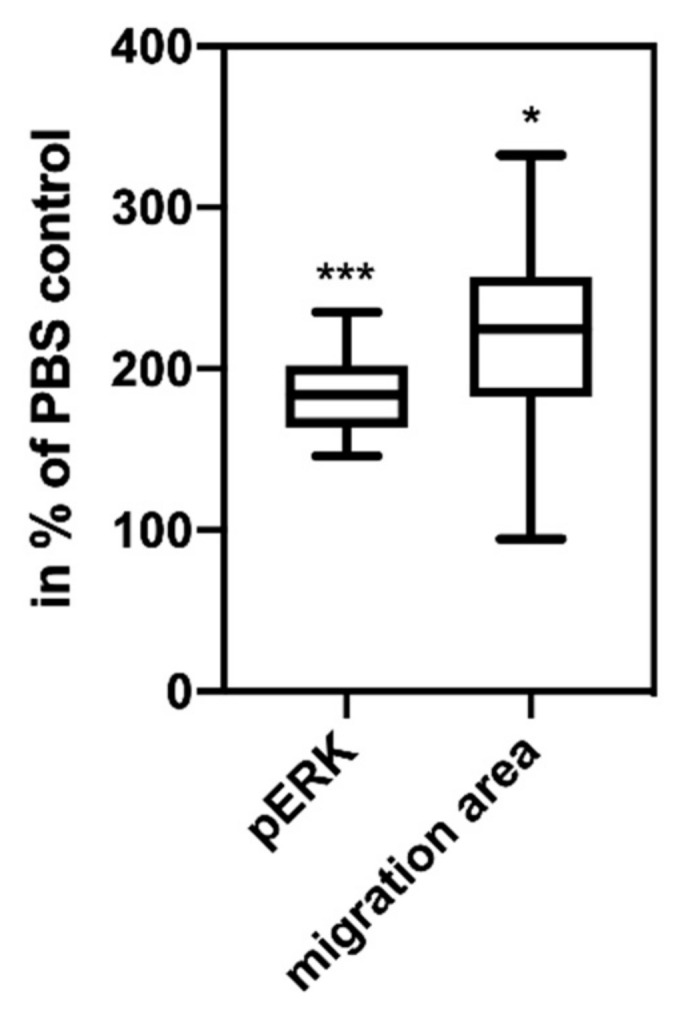
Ascites-derived EVs are biologically active and induce intracellular signaling. 10 min of co-cultivation with ascites-derived hsEVs (left box-plot, **** p* = 0.0002) and induction of tumor cell migration after 48 h of co-cultivation with ascites-derived hsEVs (right box-plot ** p* = 0.013). For these experiments, we used human pancreas carcinoma COLO 357 cells, as these cells have a stable high level of PAR2 expression, and we used them in our previous experiments in which we demonstrated the signal chain PAR2/ERK/migration [3].

## Data Availability

The data presented in this study are available on request from the corresponding author. The data are not publicly available due to anonymization of patient names.

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
