# Peer review of "Biologically Active Tissue Factor-Bearing Larger Ectosome-Like Extracellular Vesicles in Malignant Effusions from Ovarian Cancer Patients: Correlation with Incidence of Thrombosis"

_ijms, 2021, doi:10.3390/ijms22020790_

Round 1

Reviewer 1 Report

The manuscript titled “Biologically active tissue factor-bearing extracellular vesicles in malignant effusions from ovarian cancer patients: correlation with incidence of thrombosis” reports interesting pilot data, that may be considered for developing future studies with a greater sample size, on ascites from 20 patients with high-grade serous ovarian carcinoma demonstrating that microvesicles isolated from ascites provoke tumor cell migration and intracellular signaling.

Overall, I feel this manuscript is of high quality, written quite clearly and appropriate for “International Journal of Molecular Sciences”. In my opinion, the manuscript is written in good English (but I am not a native speaker, unfortunately).

This manuscript is basically publishable as is after some indispensable changes. However, only 6 minor issues need to be addressed in order to improve the quality of the paper, as discussed below:

Title:

1) I would suggest to the authors to rephrase the title replacing “extracellular vesicles” with “microvesicle fraction” or “microvesicle-enriched extracellular vesicle population” or “microvesicle fraction”. Make it unambiguous.

Abstract:

2) Line 27 of pag. 2: replace “TF+EVs” with “TF+-EVs” (superscript +). Also provide throughout the manuscript

Introduction:

3) Line 61 of pag. 2: replace “EpCAM + EVs”with “EpCAM+-EVs” (superscript +).

4) Line 73 of pag. 2: To me it’s misleading to use the term “high-speed EVs”, I'd rather use “large EVs” or or “microvesicle-enriched EVs” or at most “ectosome-like EVs”.

Material and Methods:

5) Characteristics of patients: I invite you to explain also in this subparagraph (and not only on page 7 line 241) the total number of patients enrolled.

Results:

6) Line 232 of pag. 6: you have to explain in quantitative terms what it means “considerable amounts of these markers”.

I remember that e.g. colorectal large EVs were significantly depleted of these markers (Xu, Rong, et al. "Highly-purified exosomes and shed microvesicles isolated from the human colon cancer cell line LIM1863 by sequential centrifugal ultrafiltration are biochemically and functionally distinct." Methods 87 (2015): 11-25).

Author Response

Questions: English language and style are fine/minor spell check required 

Reply: The manuscript was edited by a professional English editor.

  1. Questions: Title: I would suggest to the authors to rephrase the title replacing “extracellular vesicles” with “microvesicle fraction” or “microvesicle-enriched extracellular vesicle population” or “microvesicle fraction”. Make it unambiguous.

Re: In order to make the title and the nomenclature used throughout the manuscript as unambiguous as possible, after intense discussion in our group, we decided that the term "larger ectosome-like extracellular vesicles", represents our concept best. It would also be in accordance with the proposals of the ISEV. We have changed the title accordingly.

  1. Questions: Abstract: Line 27 of page 2: replace “TF+EVs” with “TF+-EVs” (superscript +). Also provide throughout the manuscript
  2. Introduction: Line 61 of pag. 2: replace “EpCAM + EVs” with “EpCAM+-EVs” (superscript +).

Re: We have replaced the plus sign with the superscript+. Our language editor provided us with this rule: Use a hyphen to join two or more words to form compound adjectives that precede a noun. The purpose of joining words to form a compound adjective is to differentiate the meaning from the adjectives used separately, such as up-to-date merchandise, copper-coated wire, fire-tested material, lump-sum payment, seven-year-old son, well-stocked cupboard......

There is no hyphen between an adjective – simple or compound, and the noun it modifies.

Re: Since EVs is a noun and TF+ is an adjective, we write as follows: TF+ EVs (no hyphen but a single space before EVs).  On the same principle, we write: EpCAM+ EVs (no hyphen but a single space before EVs).

  1. Questions: Line 73 of p. 2: To me it’s misleading to use the term “high-speed EVs”, I'd rather use “large EVs” or “microvesicle-enriched EVs” or at most “ectosome-like EVs”.

Re: We had an intense discussion in our group about this proposal: there is no doubt that we (the research community) do not have a method to completely separate exosomes from ectosomes. The method we use (sequential centrifugation) is widely used and probably the best we have but, unfortunately, it does not enable complete separation, but only an extensive enrichment of larger ectosomes. We believe that every working group has to prove this with appropriate experiments, just as we did (flow cytometry, nano particle tracking, antibodies plus function, such as TF-induced signaling). In order to avoid this discussion and to follow the recommended nomenclature, we would prefer to use the term "hsEVs" and explain our preference. We did this in the Introduction (last para) and the discussion (third para). This would also be in accordance with the suggestions of the ISEV.

  1. Material and Methods: Characteristics of patients: I invite you to explain also in this subparagraph (and not only on page 7 line 241) the total number of patients enrolled.

Re: Implemented

  1. Results: line 232 of p. 6: you have to explain in quantitative terms what it means “considerable amounts of these markers”.

Re: Changed to "... and found varying amounts of these markers (CD9+, 0.53-27.09%; CD63+, 1.0-12.5%; CD81, 4.18-26.25%) on the surface of hsEVs".

The debate which markers might be useful to differentiate between exosomes and ectosomes is ongoing. Unfortunately, there is no single marker that allows this distinction. In agreement with several publications, also our own previous publications, we found that  "... tetraspanin markers were not uniformly present in our preparations but on a subpopulation of CFSE+ hsEVs" (results, first para). We think that these markers need to be interpreted in the context of the other results such as CFSE positivity and flow cytometer data as we did in the discussion (3rd para).

Reviewer 2 Report

This is a small proof-of-principle study reporting biologically active tissue factor-bearing extracellular vesicles in malignant ascites from patients with high-grade serous carcinoma and correlation with incidence of thrombosis. I only have the following minor comments:

  • There is inconsistent use of terms "ovarian carcinoma" and "serous subtype" throughout the manuscript. Since all patients had high-grade serous carcinoma, please use this term introduction, perhaps abbreviate as HGSC and then use this abbreviation consistently in all sections of the manuscript.
  • Line 75: have the slides been re-reviewed for this study to confirm the diagnosis of high-grade serous carcinoma?
  • Had any of the patient received chemotherapy at the time of collecting their ascites fluid or were all 20 patients chemotherapy naive? Please clarify.

Author Response

  1. There is inconsistent use of terms "ovarian carcinoma" and "serous subtype" throughout the manuscript. Since all patients had high-grade serous carcinoma, please use this term introduction, perhaps abbreviate as HGSC and then use this abbreviation consistently in all sections of the manuscript.

Re: Implemented

  1. Line 75: have the slides been re-reviewed for this study to confirm the diagnosis of high-grade serous carcinoma?

Re: Yes, all histological diagnoses were made by the Clinic of Pathology of the UKSH Campus Luebeck. We included that information in Materials and Methods, first para.

  1. Had any of the patient received chemotherapy at the time of collecting their ascites fluid or were all 20 patients chemotherapy naive?

Re: For all but two patients, the ascites was the first symptom of their malignant disease, none of them received chemotherapy at the time of undergoing paracentesis, we included that information in the first para of Material and Methods (patient characteristics).

The manuscript was edited by a professional English editor, we marked all changes except the language corrections red in the uploaded manuscript.